# In Situ Synchrotron XRD Characterization of Piezoelectric Al_1−x_Sc_x_N Thin Films for MEMS Applications

**DOI:** 10.3390/ma16051781

**Published:** 2023-02-21

**Authors:** Wenzheng Jiang, Lei Zhu, Lingli Chen, Yumeng Yang, Xi Yu, Xiaolong Li, Zhiqiang Mu, Wenjie Yu

**Affiliations:** 1State Key Laboratory of Functional Materials for Informatics, Shanghai Institute of Microsystem and Information Technology, Chinese Academy of Sciences, Shanghai 200050, China; 2College of Materials Science and Opto-Electronic Technology, University of Chinese Academy of Sciences, Beijing 100049, China; 3Shanghai Engineering Research Center of Energy Efficient and Custom AI IC, School of Information Science and Technology, ShanghaiTech University, Shanghai 201210, China; 4School of Microelectronics, Shanghai University, Shanghai 200444, China; 5Shanghai Synchrotron Radiation Facility, Shanghai Institute of Applied Physics, Chinese Academy of Sciences, Shanghai 201204, China

**Keywords:** aluminum scandium nitride, synchrotron XRD, piezoelectric coefficient, substrate clamping effect, micro-electromechanical system

## Abstract

Aluminum scandium nitride (Al_1−x_Sc_x_N) film has drawn considerable attention owing to its enhanced piezoelectric response for micro-electromechanical system (MEMS) applications. Understanding the fundamentals of piezoelectricity would require a precise characterization of the piezoelectric coefficient, which is also crucial for MEMS device design. In this study, we proposed an in situ method based on a synchrotron X-ray diffraction (XRD) system to characterize the longitudinal piezoelectric constant d_33_ of Al_1−x_Sc_x_N film. The measurement results quantitatively demonstrated the piezoelectric effect of Al_1−x_Sc_x_N films by lattice spacing variation upon applied external voltage. The as-extracted d_33_ had a reasonable accuracy compared with the conventional high over-tone bulk acoustic resonators (HBAR) devices and Berlincourt methods. It was also found that the substrate clamping effect, leading to underestimation of d_33_ from in situ synchrotron XRD measurement while overestimation using Berlincourt method, should be thoroughly corrected in the data extraction process. The d_33_ of AlN and Al_0.9_Sc_0.1_N obtained by synchronous XRD method were 4.76 pC/N and 7.79 pC/N, respectively, matching well with traditional HBAR and Berlincourt methods. Our findings prove the in situ synchrotron XRD measurement as an effective method for precise piezoelectric coefficient d_33_ characterization.

## 1. Introduction

Recently, aluminum nitride (AlN) piezoelectric film has attracted great attention due to wireless communication technology experiencing rapid development from 2–4G to 5G [1]. Bulk acoustic wave (BAW) filters based on AlN piezoelectric film exhibit significant advantages in terms of compact size, high performance and high compatibility with the standard complementary metal-oxide-semiconductor (CMOS) process [2,3,4]. However, limited by the relatively low intrinsic longitudinal piezoelectric coefficient d_33_, the bandwidths of BAW filters based on AlN piezoelectric film are generally less than 3% [5,6]. There is an urgent need to seek breakthroughs in AlN materials. Rare earth element doping, including scandium (Sc) [7,8,9], yttrium (Y) [10] and tantalum (Ta) [11], has been proven to be an effective method to improve the piezoelectric properties of AlN materials. Among them, scandium doping is regarded as the most efficient method, since Akiyama M demonstrated a ~400% piezoelectric response increase in 2009 [7]. With the help of first-principle calculations, Tasnadi F [12] revealed that the increased intrinsic sensitivity of axial strain induced by large elastic softening leads to the anomalous enhancement of piezoelectric response by Sc introduction. For example, when the scandium concentration is 20%, the piezoelectric coefficient d_33_ of Al_0.8_Sc_0.2_N film can increase from 5.38 pC/N to 11.19 pC/N, leading to a 9% relative bandwidth of the corresponding filters [13].

It is evident that the piezoelectric response of Al_1−x_Sc_x_N film is a key parameter for the design and optimization of high-frequency acoustic resonators and filters. At present, Al_1−x_Sc_x_N growth methods include magnetron sputtering, molecular beam epitaxy (MBE) [14] and metal-organic chemical vapor deposition (MOCVD) [15]. Among them, magnetron sputtering is still the most commonly used method for Al_1−x_Sc_x_N piezoelectric material deposition due to its high compatibility with traditional device fabrication processes and relatively low manufacturing costs. Although the piezoelectric response of Al_1−x_Sc_x_N film can be predicted by theoretical calculations [12,16], the values of fabricated films often deviate from the theoretical ones, due to the deterioration of crystalline quality, intrinsic stress and calculation accuracy. In particular, with the increase in Sc concentration to around 30%, the crystal quality of Al_1−x_Sc_x_N film may significantly deteriorate because of abnormal oriented grains (AOG) formation. When Sc concentration is higher than 50%, the non-polar cubic structure is more stable than the polar wurtzite structure [16], which has adverse effects on its piezoelectric response [9,17,18,19]. Therefore, it is crucial to develop an effective and accurate experimental method for the evaluation of Al_1−x_Sc_x_N film piezoelectric property.

At present, the methods commonly used for piezoelectric characterization include PiezoMeter [8], double-beam laser interferometry (DBLI) [20], piezoresponse force microscopy (PFM) [21] and acoustic resonators [22]. d_33_ values ranging from 3.2 pC/N to 4.53 pC/N were reported by using above methods [21,23,24], while slightly higher values ranging from 4.9 pC/N to 5.1 pC/N were obtained after measurement system calibration [25,26]. However, since the piezoelectric coefficient of Al_1−x_Sc_x_N film is only tens of pC/N, about an order of magnitude smaller than barium titanate (BaTiO_3_) [27] and zirconate titanate (PZT) [28], it is very challenging to characterize Al_1−x_Sc_x_N film using these methods. Synchrotron radiation is an advanced collimated light source with a high intensity and a wide spectrum. X-ray diffraction (XRD) with an extremely powerful X-ray source produced by synchrotron radiation facilities offers a puissant and special technique by which to characterize the structure of materials on the atomic or molecular scales [29]. It provides greater accuracy to investigate material properties such as phase composition, crystallite size, strain and defect [30]. It also allows time-resolved in situ characterization with a significantly reduced measurement time and a much higher resolution, exhibiting particular advantages in structure characterization on sub-nanometer scales. For example, Shiomi et al. investigated the piezoelectric coefficient of Al_0.2_Ga_0.8_N film based on synchrotron XRD, verifying the feasibility of this method to characterize the piezoelectric response at the picometer scale [31].

Herein, in the present work, we developed an in situ measurement system based on the synchrotron radiation-powered XRD technique to characterize the piezoelectric coefficient d_33_ of Al_1−x_Sc_x_N films. The measured results quantitatively demonstrated the lattice spacing variation of Al_1−x_Sc_x_N films induced by piezoelectric effect upon applied external voltage. Longitudinal piezoelectric coefficients *d*_33_ of Al_1−x_Sc_x_N film with different Sc concentrations were successfully obtained and compared with traditional measurement methods. Moreover, by implementing substrate clamping effect calibration, the d_33_ deviation between synchrotron XRD and other methods was significantly reduced, indicating the validity of synchrotron XRD characterization for precise d_33_ measurement.

## 2. Materials and Methods

### 2.1. Synchrotron XRD Measurement System Setup

Figure 1a–c illustrates the brief fabrication process of Al_1−x_Sc_x_N samples with x = 0, 10% and 20%. Firstly, 200 nm Mo/850 nm Al_1−x_Sc_x_N/150 nm Mo sandwiched stacks were deposited by magnetron reactive sputtering on high-resistivity Si substrate with a 30 nm AlN seed layer. Those three layers were continuously deposited without a vacuum break to avoid particle or photoresist residual contamination. Highly crystalline AlN film with (0002) the full width at half maximum (FWHM) of 1.40° was observed on controlled samples, while FWHM slightly increased to 1.76° and 2.90° for Al_0.9_Sc_0.1_N and Al_0.8_Sc_0.2_N film due to the deterioration of crystallinity caused by Sc introduction, as shown in Figure 2. Then, the top Mo layer was patterned using the ion beam etching (IBE) process and acted as a hard mask for subsequent Al_1−x_Sc_x_N-selective etching. Anisotropic etching of Al_1−x_Sc_x_N films with different Sc contents was carried out by tetramethylammonium hydroxide (TMAH) for bottom mesa formation. Finally, top electrode Mo was patterned by IBE process for external voltage connection.

The schematic of the in situ synchrotron XRD measurement system is shown in Figure 1d. It consists of a high-intensity X-ray source and detector, a printed circuit board and a voltage source. The previously fabricated Al_1−x_Sc_x_N samples were mounted on the PCB board. The top and bottom electrodes were connected to the voltage source through lead bonding, which could effectively avoid the accidental penetration of piezoelectric film in the testing process. The X-ray beam was incident at the center of the top electrode to obtain symmetric diffraction as external voltage was applied to the top electrodes of samples. The shift in synchrotron XRD curves could be observed by applying different voltages. Synchrotron XRD characterization was carried out at the Shanghai Synchrotron Radiation Light Source. The line stations used were BL14B1 and BL02U2, both of which can provide X-rays with an energy of 10 keV and support the in situ piezoelectric characterization of Al_1−x_Sc_x_N samples.

### 2.2. High Overtone Bulk Acoustic Resonators Fabrication

High overtone bulk acoustic resonator (HBAR) devices, without the requirement of a suspended film structure, are widely used to estimate the piezoelectric response and electromechanical coupling coefficient with the advantage of a simplified fabrication process [32]. The fabrication process of Al_1−x_Sc_x_N HBAR devices with x = 0 and 10% is shown in Figure 3. The essential piezoelectric stacks of an HBAR device, consisting of 200 nm Mo/850 nm Al_1−x_Sc_x_N/150 nm Mo sandwiched film stack, were continuously deposited on high-resistivity Si substrate with a 30 nm AlN seed layer, as shown in Figure 3a. The top Mo and Al_1−x_Sc_x_N layers were etched by IBE and TMAH, respectively, as shown in Figure 3b. Benefiting from the highly selective and anisotropic TMAH etching, smooth and clean Al_1−x_Sc_x_N sidewalls with a 50–60° etching profile were obtained, as shown in Figure 4. Due to the lower etching rate of Al_0.9_Sc_0.1_N compared with pure AlN film, the sidewall angle of Al_0.9_Sc_0.1_N was ~10° smaller with a tapered etching profile. It should be noted that the Al_1−x_Sc_x_N film underneath the signal terminal of the top electrode should also be removed to avoid parasitic HBAR structure. Then, an oxide sidewall was formed at the boundary of the Al_1−x_Sc_x_N etching area to avoid potential short circuit risk between the top and bottom electrodes (Figure 3c). Then, 20 nm Ti/300 nm Au was deposited and accordingly patterned for pad formation. Finally, the pentagonal-shaped top electrode was patterned to define the critical piezoelectric resonance area, as shown in Figure 3e. The frequency characteristics of the reflection coefficient (S_11_) were carried out by a network analyzer KEYSIGHT E5071C.

## 3. Results and Discussion

### 3.1. d_33_ Measurement by Synchrotron XRD Method

The in situ synchrotron XRD curves of AlN, Al_0.9_Sc_0.1_N and Al_0.8_Sc_0.2_N films under different applied voltages are shown in Figure 5. In general, the diffraction peaks of the Al_1−x_Sc_x_N (0002) plane, reflecting the c-plane lattice spacing with different Sc concentrations and applied electric field, are located between 28.5° to 28.9°. As shown in Figure 5a, without external voltage, a distinct peak corresponding to AlN (0002) is located at 28.85°. A clear peak shift of 0.013° to lower angles was observed when a negative voltage of −100 V was applied; while an opposite 0.010° shift to higher angles was observed upon the application of positive 85 V. Synchrotron XRD curves of Al_1−x_Sc_x_N films with Sc concentrations of 10% and 20% exhibit similar phenomena, with peak shifts of 0.010° and 0.015° for Al_0.9_Sc_0.1_N and Al_0.8_Sc_0.2_N films under applied voltages of 60 V and 40 V separately, as shown in Figure 5b,c. It should be emphasized that thanks to the high resolution of the synchrotron radiation-powered X-ray source, diffraction peak shifts as small as 0.001° can be clearly distinguished. This behavior corresponds to the inverse piezoelectric effect of piezoelectric film, and can in turn be used to calculate the piezoelectric coefficient. By applying a longitudinal electric field E on the piezoelectric films and calculating the change rate of (0002) interplanar spacing (*d_E–_d*_0_)/*d*_0_, the longitudinal strain *S* and the piezoelectric coefficient *d*_33_ of piezoelectric films can be obtained according to Equation (1) [33]:(1)$$d_{33}={\left(\frac{\partial S_{3}}{\partial E_{3}}\right)}^{T}=\frac{\left(d_{E}-d_{0}\right)}{d_{0}}\cdot \frac{1}{E},$$
where *d_E_* and *d*_0_ are interplanar spacing with and without applied voltage. The interplanar spacing *d* is obtained by the classic Bragg formula *λ =* 2*dsinθ*, where *θ* is the position of the diffraction peak. The energy of the synchrotron radiation-powered X-ray in the work is 10 keV, corresponding to a wavelength of 0.124 nm.

The obtained XRD curves were fitted using Gaussian functions to extract the 2*θ* values for the peaks, and the c-plane lattice spacing *d = λ*/2*sinθ* and longitudinal lattice strain *S* = (*d_E_–d*_0_)/*d*_0_ were derived on the basis of the above equations. The applied electrical field dependence of the Al_1−x_Sc_x_N strain is shown in Figure 5d–f. It is worth noting that the linear relationship between the applied external voltage and the lattice strain was observed, matching very well with the theoretical piezoelectric effect of Al_1−x_Sc_x_N films. The d_33_ can be obtained by extracting the slop from the strain—electrical field correlations. The d_33_ of pure AlN film is extracted at 3.54 pm/V. When the scandium concentration increases to 10% and 20%, the d_33_ increases to 5.58 pm/V and 9.48 pm/V, corresponding to a 57.6% and a 168% improvement, respectively.

### 3.2. d_33_ Measurement by HBAR Method

As a comparison, we also adopted the commonly used acoustic resonators for d_33_ characterization. Figure 6a shows the top-view SEM image of the fabricated Al_1−x_Sc_x_N HBAR device with a pentagon-shaped resonant area. As shown in Figure 6b, the S_11_ curves of Al_1−x_Sc_x_N HBAR devices with Sc concentrations of 0% and 10% exhibit multiple resonance modes in the measured frequency range. The strongest resonance frequency is located around 3.00 GHz and 2.58 GHz, with 10.04 MHz and 9.90 MHz frequencies splitting between adjacent modes for AlN and Al_0.9_Sc_0.1_N films, respectively.

Mason models incorporated with measured scattering curves are commonly adopted for material parameter extraction [34]. The equivalent circuit of a Mason model for Al_1−x_Sc_x_N HBAR devices, consisting of an Si substrate, a bottom electrode, a piezoelectric layer and a top electrode, is shown in Figure 7; where Z, γ, C_0_ are acoustic impedance, phase shift and static capacitance, respectively. *h = e/ε^S^*, where *e* is the relevant components of the piezoelectric matrix, and *ε^S^* is the permittivity of piezoelectric materials. As shown in Figure 6c,d, the simulated results of the Mason model are in good agreement with the measured S_11_ curves. The *d*_33_ of AlN and Al_0.9_Sc_0.1_N films extracted using the Mason model are 4.22 pC/N and 6.04 pC/N, respectively.

### 3.3. d_33_ Measurement by PM300 Method

Figure 8a illustrates d_33_ measurements using the Berlincourt method PIEZOTEST PM300 [35,36]. Al_1−x_Sc_x_N piezoelectric samples were clamped between two contact probes and subjected to a low frequency force. Electrical signals generated from Al_1−x_Sc_x_N film were collected and compared with a built-in reference, and thus d_33_ values with polarization direction could be directly given by the system. As for Al_1−x_Sc_x_N sample preparation, 900 nm Al_1−x_Sc_x_N films with x = 0% and 10% were directly deposited on low resistivity Si substrates. Then, a 250 nm Au top electrode was deposited and patterned for top electrical connection, as shown in Figure 8b. The d_33_ values of AlN and Al_0.9_Sc_0.1_N films measured by PIEZOTEST PM300 were 6.33 pC/N and 8.96 pC/N, respectively.

### 3.4. d_33_ Discussion

The piezoelectric coefficient d_33_ of AlN and Al_0.9_Sc_0.1_N films measured by Synchrotron XRD, HBAR and PIEZOTEST PM300, together with theoretical calculated values, are summarized in Table 1. The theoretical calculations were performed using the Vienna Ab initio Simulation Package (VASP) [37,38,39]. It is observed that the d_33_ measured by synchrotron XRD are about 40% lower compared to that measured by PM300, while the d_33_ values obtained by synchrotron XRD and HBAR are much closer (~10%). Moreover, compared with theoretical values, the measured results from synchrotron XRD and HBAR are 22–34% smaller, while that of PM300 are 14–18% larger. One possible reason for the above deviations may come from measurement system errors, fabrication process variations and crystalline quality differences in the piezoelectric film. However, this alone is not reasonable to explain such consistently large deviations introduced by each method, and needs to be carefully considered.

Since the deformation amount of Al_1−x_Sc_x_N piezoelectric film is relatively small (only tens of picometers/voltage), the influence of the rigid substrate clamped to the piezoelectric film cannot be ignored during *d*_33_ measurement. Referring to Lefki et al. [40], the interference of the substrate during *d*_33_ measurement is quantitatively analyzed below.

The following equations can be deduced from the classic piezoelectric equation based on the symmetry of Al_1−x_Sc_x_N materials [40]:(2)$$D_{3}=d_{31}\cdot \left(T_{1}+T_{2}\right)+d_{33}\cdot T_{3}+\varepsilon_{33}^{T}\cdot E_{3},$$
(3)$$S_{1}=s_{11}^{E}\cdot T_{1}+s_{12}^{E}\cdot T_{2}+s_{13}^{E}\cdot T_{3}+d_{31}\cdot E_{3},$$
(4)$$S_{2}=s_{12}^{E}\cdot T_{1}+s_{11}^{E}\cdot T_{2}+s_{13}^{E}\cdot T_{3}+d_{31}\cdot E_{3},$$
(5)$$S_{3}=s_{13}^{E}\cdot T_{1}+s_{13}^{E}\cdot T_{2}+s_{33}^{E}\cdot T_{3}+d_{33}\cdot E_{3},$$
where $D_{3}$ is the longitudinal electrical displacement of the piezoelectric film, $S_{1},\;S_{2},\;S_{3}$ and $T_{1},\;T_{2},\;T_{3}$ are the strain and stress of the film in the in-plane x, y and out-of-plane z directions, respectively, *d*_31_ is the transverse piezoelectric coefficient, $\varepsilon_{33}^{T}$ is the dielectric constant of the film in the longitudinal direction, and $s_{11}^{E},\;s_{12}^{E},\;s_{13}^{E},\;s_{33}^{E}$ are the mechanical compliances of the piezoelectric film.

As described above, synchrotron XRD uses the inverse piezoelectric effect for *d*_33_ measurement. Ideally, the piezoelectric film would deform both in the longitudinal and in-plane directions when the voltage is applied. However, due to the restrictions from the substrate, the in-plane strains of piezoelectric film in the x and y direction *S*_1_ and *S*_2_ are assumed as zero. Since the longitudinal stress *T*_3_
*= 0*, the in-plain stress can be deduced from Equations (3) and (4) as:(6)$$T_{1}=T_{2}=\frac{-d_{31}E_{3}}{s_{11}^{E}+s_{12}^{E}}$$

The calibrated longitudinal piezoelectric coefficient *d*_33,*i*_ considering inverse piezoelectric effect can be expressed as:(7)$$d_{33,i}=\frac{\Delta l}{V}=\frac{S_{3}}{E_{3}}=d_{33,\mathrm{XRD}}+\frac{2s_{13}^{E}d_{31}}{s_{11}^{E}+s_{12}^{E}}$$
where *d*_33,XRD_ is the experimental result measured by synchrotron XRD. By using theoretical values of $s_{11}^{E},\;s_{12}^{E},s_{13}^{E}$ and *d*_31_ for a rough evaluation, the measured *d*_33,XRD_ from synchrotron XRD is underestimated because of the deformation constraint by the substrate. The *d*_33_ value of AlN and Al_0.9_Sc_0.1_N films obtained by synchronous XRD is revised to 4.76 pC/N and 7.79 pC/N according to above the correction, which is 34% and 40% larger than the originals.

The *d*_33_ extracted using the HBAR device also uses the inverse piezoelectric effect, which is similar to synchrotron XRD. The calibrated *d*_33,*r*_ can be expressed by replacing the flexibility coefficient $s_{ij}^{E}$ and piezoelectric strain coefficient $d_{ij}$ in Equation (7) with the stiffness coefficient $C_{ij}^{E}$ and piezoelectric stress coefficient $e_{ij}$, as shown below:(8)$$s_{11}^{E}=\frac{C_{11}^{E}C_{33}^{E}-C_{13}^{E}{}^{2}}{\left(C_{11}^{E}-C_{12}^{E}\right)\left[c_{33}^{E}\left(C_{11}^{E}+C_{12}^{E}\right)-2C_{13}^{E}{}^{2}\right]},$$
(9)$$s_{12}^{E}=-\frac{C_{12}^{E}C_{33}^{E}-C_{13}^{E}{}^{2}}{\left(C_{11}^{E}-C_{12}^{E}\right)\left[c_{33}^{E}\left(C_{11}^{E}+C_{12}^{E}\right)-2C_{13}^{E}{}^{2}\right]},$$
(10)$$s_{13}^{E}=-\frac{C_{13}^{E}}{c_{33}^{E}\left(C_{11}^{E}+C_{12}^{E}\right)-2C_{13}^{E}{}^{2}},$$
(11)$$d_{31}=\frac{e_{31}C_{33}^{E}-e_{33}C_{13}^{E}}{\left(C_{11}^{E}+C_{12}^{E}\right)C_{33}^{E}-2C_{12}^{E}{}^{2}},$$
(12)$$d_{33}=\frac{e_{33}\left(C_{11}^{E}+C_{12}^{E}\right)-2e_{31}C_{13}^{E}}{\left(C_{11}^{E}+C_{12}^{E}\right)C_{33}^{E}-2C_{13}^{E}{}^{2}}.$$
*d*_33,*r*_ extracted from HBAR devices can be expressed as:(13)$$d_{33,r}=\frac{e_{33}}{C_{33}^{E}}.$$

According to the above, the *d*_33,*r*_ of the AlN and Al_0.9_Sc_0.1_N films is revised to 5.43 pC/N and 8.25 pC/N, leading to a 29% and 37% increase from the originals. It is observed that the revised piezoelectric coefficient d_33_ obtained by HBAR devices is consistent with that from synchrotron XRD.

Although the Berlincourt method is the most convenient and fast way for piezoelectric coefficient measurement, the substrate would deform in-plane alongside Al_1−x_Sc_x_N film due to the Poisson effect, introducing additional in-plane strain in piezoelectric films. The relationship between strain, Poisson’s ratio *μ* and Young’s modulus of substrate *Y* can be expressed as:(14)$$S_{1}=S_{2}=-\frac{\mu T_{3}}{Y}$$

Then, the in-plane stress is straightforwardly calculated from the strain-stress equation as follows:(15)$$T_{1}=T_{2}=\frac{-\frac{\mu T_{3}}{Y}-d_{31}E_{3}-s_{13}^{E}T_{3}}{s_{11}^{E}+s_{12}^{E}}$$

The calibrated longitudinal piezoelectric coefficient *d*_33,*d*_ considering the substrate-restricted direct piezoelectric effect can be expressed as:(16)$$d_{33,d}=\frac{Q}{F_{3}}=\frac{D_{3}}{T_{3}}=d_{33,\mathrm{PM}300}+2d_{31}\frac{\frac{\mu }{Y}+s_{13}^{E}}{s_{11}^{E}+s_{12}^{E}},$$
where *d*_33,PM300_ is the experimental result measured by PM300. $2d_{31}\frac{\frac{\mu }{Y}+s_{13}^{E}}{s_{11}^{E}+s_{12}^{E}}$ in Equation (16) is negative for Al_1−x_Sc_x_N film due to additional in-plane stress, leading to an overestimated *d*_33,PM300_ measurement results. The *d*_33_ of AlN and Al_0.9_Sc_0.1_N films obtained by PM300 is thus revised to 4.29 pC/N and 7.03 pC/N from 6.33 pC/N and 8.96 pC/N. It should be noted that the actual measurement process is much more complex due to the influence of the substrate’s resistivity, quasi-static external force and fixed clamping force during PM300 measurement.

Table 1 summarizes the d_33_ results of synchrotron XRD, PM300 and HBAR before and after substrate-related calibration. The revised d_33_ value of synchrotron XRD is 34–40% higher, while the value of PM300 is 22–32% lower compared with the original values. As a result, the deviation obtained by these two methods is reduced from up to 70% to about 10%. The deviation between the synchrotron XRD and HBAR results is within 15%, and the deviation between synchrotron XRD and theoretical values is within 13%, which is believed to be caused by the crystalline quality, sample structure and measurement configurations [41]. Moreover, the d_33_ value of Al_0.9_Sc_0.1_N films is consistent in the range of 7.03 pC/N to 8.25 pC/N obtained by different methods, which is always 52–64% higher than pure AlN. This is consistent with previous reports [8,16,42], indicating that Sc introduction can significantly improve the intrinsic piezoelectric property of AlN film. The d_33_ values obtained using the synchrotron XRD method are also in good agreement with previously reported results, as shown in Table 2. The above results demonstrate the feasibility and effectiveness of the synchrotron XRD method and the applicability of the proposed revised models.

The Berlincourt method is predominantly used for d_33_ measurement due to its simple and quick measurement process. However, it is very difficult to produce a homogeneous uniaxial stress on the piezoelectric film, and the results may be affected by the sample clamping situation and electrode size; therefore, the method is usually suggested for d_33_ relative comparison [48]. The HBAR method can be not only used for d_33_ measurement, but also for a complete set of material parameters, such as the dielectric constant, sound velocity and density [49]. However, it requires a complete fabrication process of acoustic devices, and the results strongly depend on the accuracy of the resonance frequencies and the fitting precision of a large group of material parameters. Compared with the HBAR method, the sample fabrication process of synchrotron XRD is much simpler. What is more, though a few theoretical parameters are used during substrate clamping effect calibration, the d_33_ is directly extracted from the strain—electric field correlation without the involvement of material parameters, and the accuracy of the obtained results can also be roughly verified by the linearity of the strain—electric field relation.

## 4. Conclusions

In summary, synchrotron XRD, together with HBAR devices and a PM300 PiezoMeter, was adopted to accurately characterize the longitudinal piezoelectric coefficient (d_33_) of Al_1−x_Sc_x_N films in this paper. By considering the substrate clamping effect, more reliable and consistent d_33_ results were obtained, exhibiting the indispensability of the correction metlongitudinal piezoelectric coefficient hod for piezoelectric characterization, especially for materials with a small piezoelectric coefficient, such as the Al_1−x_Sc_x_N system. Moreover, synchrotron XRD was proven as an effective method by which to evaluate the accuracy of d_33_ extracted from PIEZOTEST PM300 and HBAR, providing a feasible, precise and alternative technique for piezoelectric response characterization.

## Figures and Tables

**Figure 1 materials-16-01781-f001:**
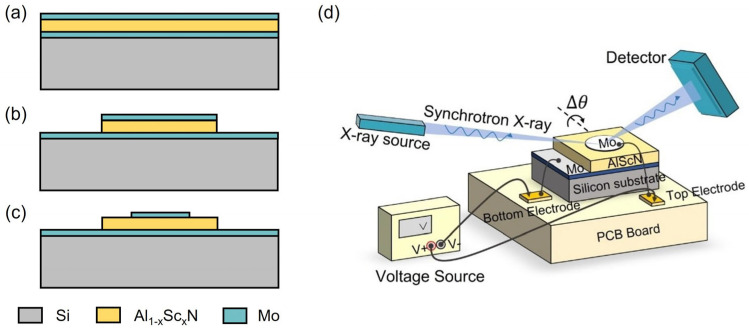
(**a**) Mo/Al_1−x_Sc_x_N/Mo piezoelectric stack deposition; (**b**) Top Mo/Al_1−x_Sc_x_N etching; (**c**) Top electrode Mo patterning; (**d**) Schematic of the measurement setup for the in situ synchrotron XRD characterization under static voltage supply.

**Figure 2 materials-16-01781-f002:**
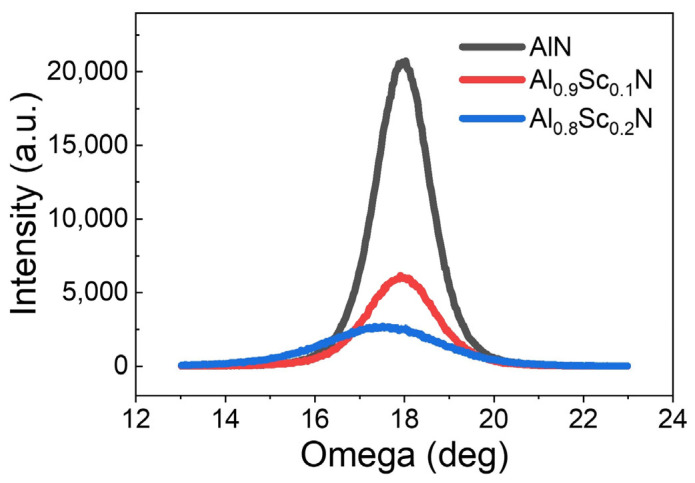
The XRD rocking curves of (0002) plane for AlN, Al_0.9_Sc_0.1_N and Al_0.8_Sc_0.2_N.

**Figure 3 materials-16-01781-f003:**
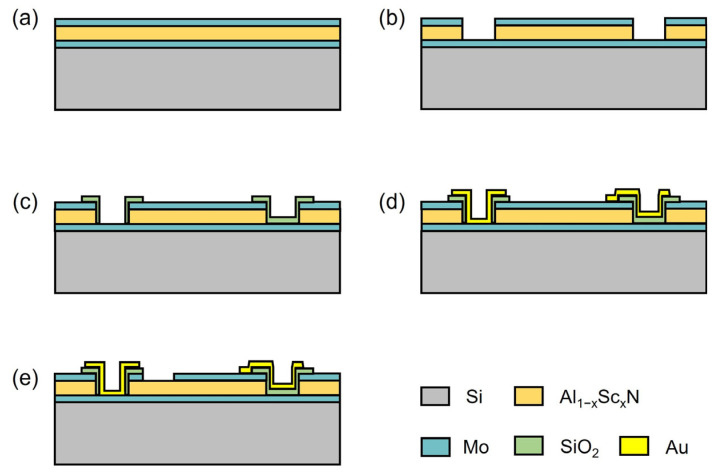
The fabrication process of Al_1−x_Sc_x_N HBAR devices. (**a**) Mo/Al_1−x_Sc_x_N/Mo piezoelectric stack deposition, (**b**) Top Mo/Al_1−x_Sc_x_N patterning, (**c**) SiO_2_ isolation layer deposition and patterning, (**d**) Au contact layer deposition and patterning, (**e**) Top electrode Mo patterning.

**Figure 4 materials-16-01781-f004:**
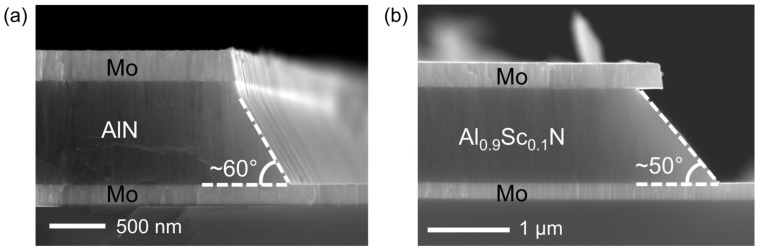
Cross-section SEM images of (**a**) AlN and (**b**) Al_0.9_Sc_0.1_N piezoelectric stacks.

**Figure 5 materials-16-01781-f005:**
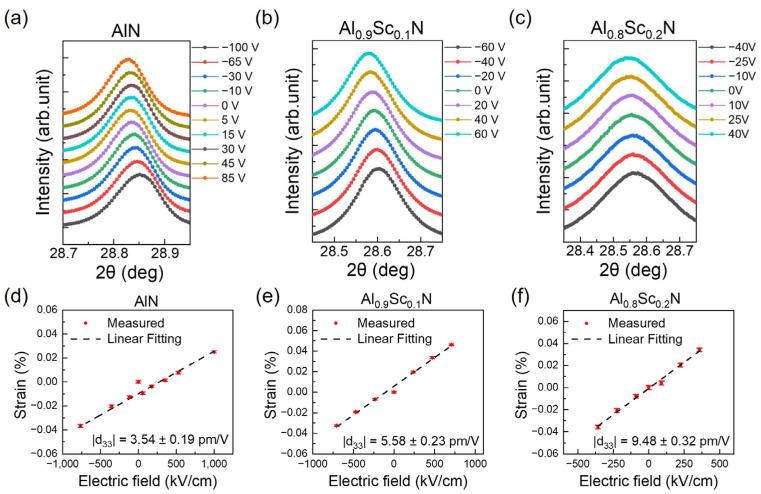
Synchrotron XRD curves of (**a**) AlN, (**b**) Al_0.9_Sc_0.1_N, (**c**) Al_0.8_Sc_0.2_N under different applied voltages V; Electric field E dependence of longitudinal strains of (**d**) AlN, (**e**) Al_0.9_Sc_0.1_N, (**f**) Al_0.8_Sc_0.2_N derived from synchrotron XRD peaks. The data from the standard PDF cards for AlN (No. 00-025-1133) are referred.

**Figure 6 materials-16-01781-f006:**
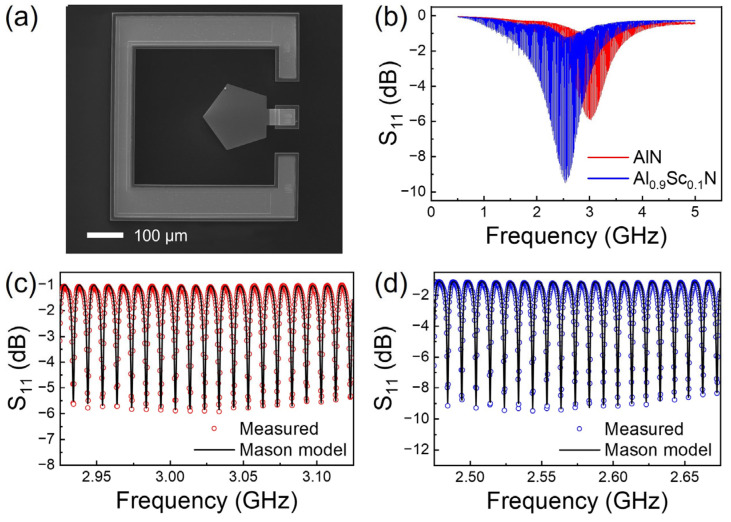
(**a**) Top-view SEM image of Al_1−x_Sc_x_N HBAR device; (**b**) S_11_ response of HBAR devices with AlN and Al_0.9_Sc_0.1_N films; Mason equivalent circuit fitting of (**c**) AlN and (**d**) Al_0.9_Sc_0.1_N on a 200 MHz span.

**Figure 7 materials-16-01781-f007:**
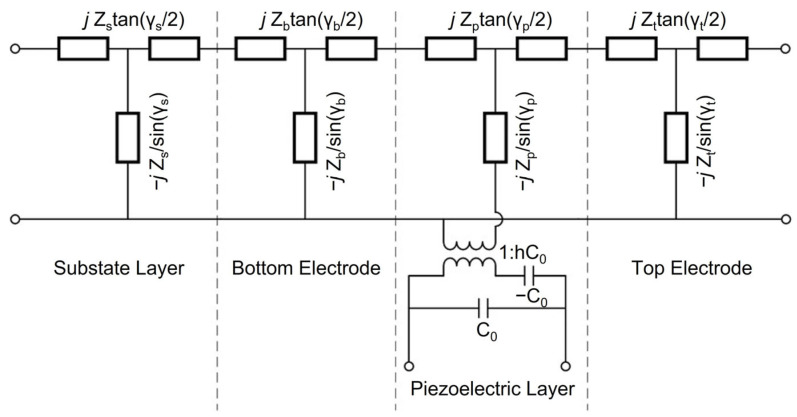
Equivalent circuit of the Mason model for Al_1−x_Sc_x_N HBAR device.

**Figure 8 materials-16-01781-f008:**
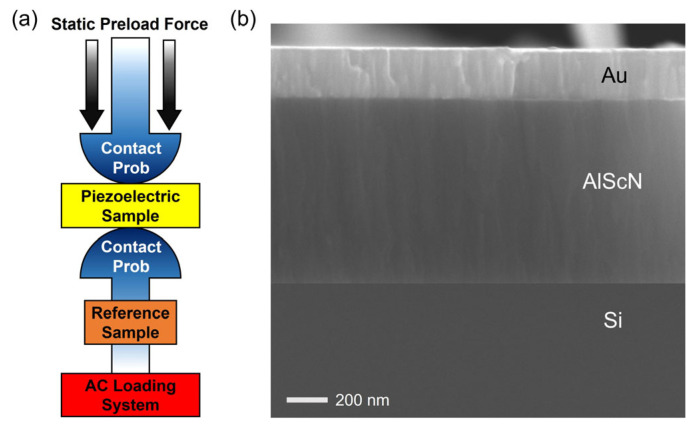
(**a**) Schematic of d_33_ measurement by PIEZOTEST PM300 [36]; (**b**) Cross-section SEM image of piezoelectric sample.

**Table 1 materials-16-01781-t001:** The measured and corrected d_33_ of AlN and Al_0.9_Sc_0.1_N films from synchrotron XRD, PM300, HBAR and theoretical calculation.

Material	Method	Measured d_33_ (pC/N)	Corrected d_33_ (pC/N)	Theoretical Calculated d_33_ (pC/N)
AlN	Synchrotron XRD	3.54	4.76	5.38
	PM300	6.33	4.29	
	HBAR	4.22	5.43	
Al_0.9_Sc_0.1_N	Synchrotron XRD	5.58	7.79	7.89
	PM300	8.96	7.03	
	HBAR	6.04	8.25	

**Table 2 materials-16-01781-t002:** d_33_ comparison with other reports.

Material	d_33_ (pC/N)	Reference	Material	d_33_ (pC/N)	Reference
AlN	4.53	[24]	Al_0.9_Sc_0.1_N	7.5	[8]
AlN	4.76	Our work	Al_0.9_Sc_0.1_N	7.79	Our work
AlN	4.9	[25]	Al_0.88_Sc_0.12_N	7.9	[43]
AlN *	5.0	[44]	Al_0.85_Sc_0.15_N	7.92	[22]
AlN	5.1	[26]	Al_0.83_Sc_0.17_N	9.5	[43]
AlN *	5.1	[45]	Al_0.8_Sc_0.2_N	11.5	[8]
AlN	5.53	[46]	Al_0.75_Sc_0.25_N	13.2	[47]

* Obtained by theoretical calculations.

## Data Availability

The data presented in this study are available on request from the corresponding author.

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
