# Peer review of "In Situ Synchrotron XRD Characterization of Piezoelectric Al1−xScxN Thin Films for MEMS Applications"

_materials, 2023, doi:10.3390/ma16051781_

Round 1
Reviewer 1 Report
The article entitled “In situ Synchrotron XRD Characterization of piezoelectric Al1-xScxN thin films for MEMS applications” describes how to obtain the d33 piezoelectric coefficient of Al1-xScxN thin films using the synchrotron. The results are compared with the usually used HBAR and PM300 method. Synchrotron and HBAR give similar results.
The article is in general well presented and the figure are of high quality but the gain in using the synchrotron compared to the HBAR method is not discuss in paragraph 3.4 and by consequence not clear to me. Both methods need lots of mathematical corrections that in turns need the use of several theoretical values (which can be wrong according to table 1). HBAR has the advantages that can be done in-house limiting the cost. Could be this point detailed more? Maybe in the discussion or in the conclusion giving more future perspective on why the community should consider this new method?
In section 3.1 it would be good to details with one additional formula how S (strain) of graph 5 d-e-f is calculated.
In table 1 I am confused on the reported values for XRD and HBAR method for both samples. For example for HBAR - AIN sample in section 3.2 a value of 4.14 is reported while in the table I find a value of 4.22. which one is the correct? The “corrected method” was calculated using which of the two results?
Author Response
Dear Reviewer:
We would like to thank you for critical reading of the manuscript and your valuable insights which will improve the quality of our manuscript. We have carefully revised the manuscript accordingly and the following is our reply. We hope our responses listed below bring more clarifications and make our manuscript suitable for publication in “Materials”.
Yours sincerely,
Authors
Comment 1: The article is in general well-presented and the figure are of high quality but the gain in using the synchrotron compared to the HBAR method is not discuss in paragraph 3.4 and by consequence not clear to me. Both methods need lots of mathematical corrections that in turns need the use of several theoretical values (which can be wrong according to table 1). HBAR has the advantages that can be done in-house limiting the cost. Could be this point detailed more? Maybe in the discussion or in the conclusion giving more future perspective on why the community should consider this new method?
Our response: We would like to thank the reviewer for your positive comments on our work. It is true that HBAR method shows some advantages for limited cost and resources, leading to become one of the mainstream methods for piezoelectric film d33 characterization. However, the fabrication process of HBAR device is more complicated compared with synchrotron XRD method, which may introduce more variation and noise for d33 extraction. What’s more, the accuracy of d33 characterization using HBAR is largely dependent on the precision of the Mason model fitting. However, several material parameters, including dielectric constant, sound velocity, density, stiffness and acoustic loss, should be considered during the fitting process. Those parameters may vary with film deposition conditions and device fabrication processes, introducing further uncertainty for d33 extraction.
Regarding d33 characterization by synchrotron XRD method, on the one hand, the sample fabrication process is much simpler, which helps minimize process-related impacts. On the other hand, though the theoretical values of mechanical compliances and transverse piezoelectric coefficient d31 were used during substrate clamping effect calibration, the original value of d33 was extracted from the strain-electric field correlation, in which the strain is directly calculated from the XRD curves, and electric field is applied by external voltage source, neither need involve material parameters as above. What’s more, the rationality and accuracy of the results can also be roughly verified by the linearity between strain and electric field correlation.
The below content was added at Section 3.4 for better clarifications:
The Berlincourt method is predominantly used for d33 measurement due to its simple and quick measurement process. However, it is very difficult to produce a homogeneous uniaxial stress on the piezoelectric film and the results may be affected by the sample clamping situation and electrode size, therefore, the method is usually suggested for d33 relative comparison[48]. HBAR method can be used not only for d33 measurement, but also for a complete set of material parameters, such as dielectric constant, sound velocity and density[49]. However, it requires a complete fabrication process of acoustic devices, and the results strongly de-pend on the accuracy of the resonance frequencies and the fitting precision of a large group of material parameters. Compared with HBAR method, the sample fabrication process of synchrotron XRD is much simpler. What’s more, though a few theoretical parameters were used during substrate clamping effect calibration, the d33 is extracted directly from the strain - electric field correlation without the involvement of material parameters, and the accuracy of the obtained results can also be roughly verified by the linearity of strain - electric field relation.
Corresponding change in manuscript: Yes
Location of Change:
Section: 3.4. d33 Discussion
Page 10, row 310
Comment 2: In section 3.1 it would be good to details with one additional formula how S (strain) of graph 5 d-e-f is calculated.
Our response: As the reviewer suggested, the c-plane lattice spacing d = λ / 2sinθ and longitudinal lattice strain S = (dE - d0) / d0 were added in revised manuscript for better clarification.
The obtained XRD curves were fitted using Gaussian functions to extract the 2θ values for the peaks, and the c-plane lattice spacing and longitudinal lattice strain were derived on the basis of above equations.
The above sentence was modified as follows:
The obtained XRD curves were fitted using Gaussian functions to extract the 2θ values for the peaks, and the c-plane lattice spacing d = λ / 2sinθ and longitudinal lattice strain S = (dE - d0) / d0 were derived on the basis of above equations.
Corresponding change in manuscript: Yes
Location of Change:
Section: 3.1. d33 Measurement by Synchrotron XRD Method
Page 5, row 178
Comment 3: In table 1 I am confused on the reported values for XRD and HBAR method for both samples. For example for HBAR - AIN sample in section 3.2 a value of 4.14 is reported while in the table I find a value of 4.22. which one is the correct? The “corrected method” was calculated using which of the two results?
Our response: Thank the reviewer for pointing this out. We have made the correction according to the reviewer’s comments, and the whole paper has been carefully proofread for several times.
The d33 of AlN and Al0.9Sc0.1N films extracted from Mason model is 4.14 pC/N and 6.11 pC/N, respectively.
The above sentence was modified as follows:
The d33 of AlN and Al0.9Sc0.1N films extracted from Mason model is 4.22 pC/N and 6.04 pC/N, respectively.
Moreover, compared with theoretical values, the measured results from synchrotron XRD and HBAR are 22% - 30% smaller, while that of PM300 are 14% - 18% larger.
The above sentence was modified as follows:
Moreover, compared with theoretical values, the measured results from synchrotron XRD and HBAR are 22% - 34% smaller, while that of PM300 are 14% - 18% larger.
Corresponding change in manuscript: Yes
Location of Change:
Section: 3.2. d33 Measurement by HBAR Method;
3.4. d33 Discussion
Page 7, row 211; Page 8, row 236
Table 1
Please see the attachment.

Reviewer 2 Report
The authors reported very interesting results of Aluminum scandium nitride (Al1-xScxN) film as a piezoelectric response material for micro-electromechanical system (MEMS) applications.
I suggest publishing this study after some major corrections:
· May you add some of your in-situ synchrotron XRD results in the abstract
· May you check the reference format by considering the format of the Materials journal
· Did you consider your angle from the x-ray source to the detector? which one is constant during the measurements
· Please can you add the reference of equation 1
· I see that the literature review should be added in detail to make a good comparison with your results
· What will be the exact effects of increasing the ratio of Sc larger than 0.2
· Which software did you use to measure the Theoretical part
· I think your results are interesting but please can you make a good comparison of Al1-2 xScxN with the reported materials
Author Response
Dear Reviewer:
We would like to thank you for critical reading of the manuscript and your valuable insights which will improve the quality of our manuscript. We have carefully revised the manuscript accordingly and the following is our reply. We hope our responses listed below bring more clarifications and make our manuscript suitable for publication in “Materials”.
Yours sincerely,
Authors
Comment 1: The authors reported very interesting results of Aluminum scandium nitride (Al1-xScxN) film as a piezoelectric response material for micro-electromechanical system (MEMS) applications. I suggest publishing this study after some major corrections.
Our response: We would like to thank the reviewer for your positive comments on our work.
Comment 2: May you add some of your in-situ synchrotron XRD results in the abstract
Our response: Thanks for the reviewer’s suggestion. d33 results obtained by synchrotron XRD method was added in the abstract of revised manuscript for better clarification.
The below content was added at Abstract:
The d33 of AlN and Al0.9Sc0.1N obtained by synchronous XRD method is 4.76 pC/N and 7.79 pC/N, respectively, matches well with traditional HBAR and Berlincourt methods.
Corresponding change in manuscript: Yes
Location of Change:
Section: Abstract
Page 1, row 28
Comment 3: May you check the reference format by considering the format of the Materials journal
Our response: Thank the reviewer for point this out. The format of the references has been updated following Materials journal guideline.
Corresponding change in manuscript: Yes
Location of Change:
Section: References
Page 11-14
Comment 4: Did you consider your angle from the x-ray source to the detector? which one is constant during the measurements
Our response: Thank the reviewer for pointing this out. θ-2θ mode was used in our Synchrotron XRD method. The X-ray source is fixed, and the sample and detector are rotated synchronously. It is worth noting that the energy of synchrotron-powered X-ray in our work is 10 keV, corresponding to the wavelength λ of 0.124 nm, therefore, the diffraction peak of AlN (0002) located at about 28° according to the Bragg formula λ = 2dsinθ. However, the wavelength λ of traditional X-ray source is generally 0.154 nm, corresponding to the diffraction peak about 18°. So the diffraction peaks measured from our XRD system are different with previous reports.
Corresponding change in manuscript: No
Comment 5: Please can you add the reference of equation 1
Our response: The reference has been added in revised manuscript according to the reviewer’s suggestion.
Reference 33 as below was added at Reference:
33. IEEE Standard on Piezoelectricity. ANSI/IEEE Std 176-1987 1988; doi: 10.1109/IEEESTD.1988.79638.
Corresponding change in manuscript: Yes
Location of Change:
Section: References
Page 13, row 425
Comment 6: I see that the literature review should be added in detail to make a good comparison with your results
Our response: According to the reviewer’s suggestion, previous reports on the d33 measurement results of AlN piezoelectric film have been added in the Introduction part for better clarification.
The below content was added at Abstract part:
The d33 values ranging from 3.2 pC/N to 4.53 pC/N were reported by using above methods[21,23,24], while slightly higher values ranging from 4.9 pC/N to 5.1 pC/N were obtained after measurement system calibration[25,26].
Corresponding change in manuscript: Yes
Location of Change:
Section: 1. Introduction
Page 2, row 70
Comment 7: What will be the exact effects of increasing the ratio of Sc larger than 0.2
Our response: Currently, Sc introduction is regarded as the most efficient method to increase the piezoelectric response of wurtzite-type AlN film, and extensive experimental and computational work has been carried out on continuous increasing of Al1-xScxN piezoelectric response. The first-principles calculations revealed that the piezoelectric response improvement is mainly caused by the large elastic soften along c-axis and significant intrinsic sensitivity increasing to axial strain as Sc concentration x increases from 0 to 0.75. However, the energy stability calculation shows that the polar wurtzite-type structure would convert to non-polar cubic-type structure when x exceeds about 0.5, which may be the mainly reason for the observed decrease of the piezoelectric constants of Al1-xScxN film at x around 0.43 experimentally. Moreover, as the Sc concentration x increased above 0.2, abnormal oriented grains (AOG) were observed, which is believed caused by the Sc segregation and secondary nucleation at grain boundaries. The AOG exhibit partial or total loss of c-axis texture, which has adverse effects on its piezoelectric response. So, the deposition process should be carefully optimized to suppress the AOG and crystal structure transformation for Al1-xScxN piezoelectric film fabrication with x above 0.2.
Corresponding change in manuscript: No
Comment 8: Which software did you use to measure the Theoretical part
Our response: In our manuscript, the first-principles calculations were performed using the Vienna Ab initio Simulation Package (VASP) software.
The below content was added at revised manuscript:
The theoretical calculations were performed using the Vienna Ab initio Simulation Package (VASP) [37-39].
Corresponding change in manuscript: Yes
Location of Change:
Section: 3.4. d33 Discussion
Page 8, row 231
Comment 9: I think your results are interesting but please can you make a good comparison of Al1- xScxN with the reported materials
Our response: Thanks for the reviewer’s suggestion. Previously reported results are summarized and presented in Table 2 of the revised manuscript for better comparison.
The below content was added at revised manuscript:
The d33 values obtained by synchrotron XRD method are also in good agreement with previously reported results, as shown in Table 2.
Table 2. d33 comparison with other reports.
|
Material |
d33 (pC/N) |
Reference |
Material |
d33 (pC/N) |
Reference |
|
AlN |
4.53 |
[24] |
Al0.9Sc0.1N |
7.5 |
[8] |
|
AlN |
4.76 |
Our work |
Al0.9Sc0.1N |
7.79 |
Our work |
|
AlN |
4.9 |
[25] |
Al0.88Sc0.12N |
7.9 |
[43] |
|
AlN* |
5.0 |
[44] |
Al0.85Sc0.15N |
7.92 |
[22] |
|
AlN |
5.1 |
[26] |
Al0.83Sc0.17N |
9.5 |
[43] |
|
AlN* |
5.1 |
[45] |
Al0.8Sc0.2N |
11.5 |
[8] |
|
AlN |
5.53 |
[46] |
Al0.75Sc0.25N |
13.2 |
[47] |
* Obtained by theoretical calculations
Corresponding change in manuscript: Yes
Location of Change:
Section: 3.4. d33 Discussion
Page 10, row 303
Please see the attachment.

Reviewer 3 Report
Dear Authority,
The manuscript entitled ‘In-situ Synchrotron XRD Characterization of Piezoelectric Al1-xScxN Thin Films for MEMS Applications’ presents different measurement methods for the longitudinal piezoelectric coefficient (d33) such as synchrotron XRD, the conventional high over-tone bulk acoustic resonators (HBAR) devices and Berlincourt methods. The data extracted from these methods is not coherent with the theoretical calculation. Such a differences in longitudinal piezoelectric coefficient (d33) measurement is attributed to measurement system errors, fabrication process variation and crystalline quality difference of the piezoelectric film. Further calculations are applied for correction.
I think, the xrd measurement using synchrotron radiation could be useful for measurement of longitudinal piezoelectric coefficient (d33) in MEMS. So, the study could be valuable for literature after modifications.
1- The PDF numbers for the AlN phase needs to be provided in body of the manuscript
2- The XRD measurement is done through Mo electrode which eventually decreases the quality of data acquired from the Al1-xScxN sample under Mo electrode. I would recommend transmission mode XRD analysis if it is possible in the synchrotron facility.
3- The lattice strain calculated from peak shift on 002 reflection represents linear response of lattice under different voltages. The piezoelectric coefficient calculated from this data represents linear behavior so that there is huge differences on measured and calculated results. Therefore, the lattice strain needs to calculate from the changes on unit cell volume. Please consider following paper (https://doi.org/10.1016/j.mtcomm.2022.104272). In my opinion, there are two restrictions for this measurement. First, the xrd data represented rocking curves, is it possible to take whole spectrum for unit cell volume calculation. Second, if the thin film deposited on 0002 orientation, you could probably not see other peak reflections properly. Please revisit the experimental setup according to my comments
4- After correction, the corrected and theoretical calculated data are still not converged closely. I think you need go over the calculation and fittings on xrd data. For instance, the crystalline structure to amorphous structure huge effect on the piezoelectric coefficient (d33), the addition of Sc element in system leads huge amorphization or transition from long range order to short range order. Please visit following paper ( https://doi.org/10.1115/1.4006881)
5- The lattice strain calculated from peak shifting could be correct until the system is distorted elastically. If the system is deformed plastically, the peak shifting is constrained and different behavior on peak reflection is aroused such as peak broadening and reduction on peak intensity. Therefore, please make sure if you deformed the system elastically under different voltages. Otherwise, the measurement and the calculation becomes all wrong and misleading. For more information please consider following paper in your manuscript (https://doi.org/10.31590/ejosat.878002
I believe the manuscript needs be modification before publication in Materials
Best wishes,

Author Response
Dear Reviewer:
We would like to thank you for critical reading of the manuscript and your valuable insights which will improve the quality of our manuscript. We have carefully revised the manuscript accordingly and the following is our reply. We hope our responses listed below bring more clarifications and make our manuscript suitable for publication in “Materials”.
Yours sincerely,
Authors
Comment 1: I think, the xrd measurement using synchrotron radiation could be useful for measurement of longitudinal piezoelectric coefficient (d33) in MEMS. So, the study could be valuable for literature after modifications.
Our response: We would like to thank the reviewer for your positive comments on our work.
Comment 2: The PDF numbers for the AlN phase needs to be provided in body of the manuscript
Our response: Thank the reviewer for pointing this out. The PDF number of AlN is added in revised manuscript according to the reviewer’s suggestion.
The below content was added at the caption of Figure 5:
The data from the standard PDF cards for AlN (No. 00-025-1133) is referred.
Corresponding change in manuscript: Yes
Location of Change:
Section: 3.1. d33 Measurement by Synchrotron XRD Method
Page 6, row 191
Comment 3: The XRD measurement is done through Mo electrode which eventually decreases the quality of data acquired from the Al1-xScxN sample under Mo electrode. I would recommend transmission mode XRD analysis if it is possible in the synchrotron facility.
Our response: Thank the reviewer for pointing this out. Unfortunately, the XRD facility we used doesn’t support transmission mode. However, the measurement accuracy of synchrotron XRD with traditional reflection mode has been well demonstrated in previous reports (Scientific Reports, 9, 7309, 2019; Applied Physics Express, 14, 095502, 2021).
Corresponding change in manuscript: No
Comment 4: The lattice strain calculated from peak shift on 002 reflection represents linear response of lattice under different voltages. The piezoelectric coefficient calculated from this data represents linear behavior so that there is huge differences on measured and calculated results. Therefore, the lattice strain needs to calculate from the changes on unit cell volume. Please consider following paper (https://doi.org/10.1016/j.mtcomm.2022.104272). In my opinion, there are two restrictions for this measurement. First, the xrd data represented rocking curves, is it possible to take whole spectrum for unit cell volume calculation. Second, if the thin film deposited on 0002 orientation, you could probably not see other peak reflections properly. Please revisit the experimental setup according to my comments
Our response: Thank the reviewer for sharing the ideas and thorough analysis, and the recommended paper was read carefully, it is an interesting and useful method to reveal ceramic structural alteration by unit cell volume. However, different from the reference paper of millimeter-thick 8%YSZ characterization with transmission measurement mode, synchrotron XRD method with reflection mode was used in our work to characterize sub-micrometer-thick Al1-xScxN piezoelectric films, which were deposited on ~500 μm thick Si substrate. When an external voltage is applied on the sample, interplanar spacings along the out-of-plane direction change accordingly due to the piezoelectric effect. However, the interplanar spacings along in-plane direction were restricted from the ~500 μm thick Si substrate, leading to the in-plane strain of Al1-xScxN film in the x and y direction S1 = S2 = 0. Based on the piezoelectric equation (5) in manuscript, the below equation can be deduced:

(The original formula could be found in the attachment.)
where S3 and E3 are strain and electric field of the film in the out-of-plane z direction. sijE, d31 and d33 are the mechanical compliances, transverse and longitudinal piezoelectric coefficients, which are all the intrinsic material parameters of Al1-xScxN films. The equation above exhibits a linear behavior between strain and electric field, which matches reasonably well with our experimental results, as shown in manuscript Figure 5(d)-(f). Moreover, considering the substrate clamping effect, the in-plane interplanar spacing variation under different external voltages is assumed as 0, therefore, the interplanar spacing change in the longitudinal direction is equal to the change in unit cell volume. Based on the discussion above, d33 characterization by our synchrotron XRD method is reasonably accurate. In addition, similar synchrotron XRD methods have been demonstrated the feasibility in previous reports for lead zirconate titanate (PZT) and AlGaN/GaN piezoelectric films (Scientific Reports, 9, 7309, 2019; Applied Physics Express, 14, 095502, 2021), and those results are in good agreement with conventional methods.
Corresponding change in manuscript: No
Comment 5: After correction, the corrected and theoretical calculated data are still not converged closely. I think you need go over the calculation and fittings on xrd data. For instance, the crystalline structure to amorphous structure huge effect on the piezoelectric coefficient (d33), the addition of Sc element in system leads huge amorphization or transition from long range order to short range order. Please visit following paper (https://doi.org/10.1115/1.4006881)
Our response: We agree with the reviewer’s comment. Indeed, the d33 of Al1-xScxN extracted from the calibrated Synchrotron XRD method is still 1%-13% lower compared with theoretical values. Except the variation from the measurement system and device fabrication deviation, the crystalline quality and sample structure parameters can also considerably affect the measurement results, which was thoroughly investigated in the recommended reference by using PZT piezoelectric films as an example. As demonstrated in the reference, the d33 obtained by the new approach still shows 14%-32% deviation from the manufacturer’s specifications after finite element model calibration.
In our work, inverse piezoelectric effect is utilized for d33 characterization. Non-destructively external electric field was applied on the sample through metal lead and electrodes, which can avoid non-uniformed mechanical force and potential sample damage. However, polycrystalline Al1-xScxN piezoelectric films were deposited by magnetron reactive sputtering, with (0002) the full width at half maximum (FWHM) ranging from 1.4° - 2.9°. What’s more, as Sc concentration increasing, abnormal oriented grains (AOG) would form during sputtering process, which is believed caused by the Sc segregation and secondary nucleation at grain boundaries. The AOG exhibit partial or total loss of c-axis texture, which has adverse effects on its piezoelectric response. Therefore, the crystalline quality of Al1-xScxN films would largely contribute the deviations between the measured and theoretical values.
The deviation between synchrotron XRD and HBAR results is within 15%, indicating the effectiveness of synchrotron XRD method and the applicability of the proposed revised models.
The above sentence was modified as follows:
The deviation between synchrotron XRD and HBAR results is within 15%, and the deviation between synchrotron XRD and theoretical values is within 13%, which is believed caused by the crystalline quality, sample structure and measurement configurations[41] ... Above results demonstrated the feasibility and effectiveness of synchrotron XRD method and the applicability of the proposed revised models.
Corresponding change in manuscript: Yes
Location of Change:
Section: 3.4. d33 Discussion
Page 10, row 297
Comment 6: The lattice strain calculated from peak shifting could be correct until the system is distorted elastically. If the system is deformed plastically, the peak shifting is constrained and different behavior on peak reflection is aroused such as peak broadening and reduction on peak intensity. Therefore, please make sure if you deformed the system elastically under different voltages. Otherwise, the measurement and the calculation become all wrong and misleading. For more information please consider following paper in your manuscript (https://doi.org/10.31590/ejosat.878002)
Our response: Thank the reviewer for your kind reminder. The Al1-xScxN samples were confirmed as elastic distortion within the measurement range. In our measurement, the applied voltage was scanned from negative to positive direction. For example, the external voltages of -60 V, -40 V, -20 V, 0 V, 20 V, 40 V and 60 V were sequentially applied on Al0.9Sc0.1N sample. As shown in manuscript Figure 5(d)-(f), symmetric diffraction peak shifts were observed for both negative to 0 V and 0 V to positive voltage ranges for Al1-xScxN samples with x = 0, 0.1, 0.2, indicating the elastic distortion of Al1-xScxN samples during the measurement. What’s more, per reviewer suggested, the (0002) the full width at half maximum (FWHM) and peak intensity of AlN and Al0.9Sc0.1N under different applied voltages were shown in Figure.S1 as below. As the applied voltages scanned from negative to positive, FWHM and peak intensity of Al1-xScxN (0002) keep reasonably stable, further proves the elastic deformation of the sample during synchrotron XRD measurement.

Figure.S1 The FWHM and peak intensity of AlN and Al0.9Sc0.1N under different applied voltage
Corresponding change in manuscript: No
Please see the attachment.

Round 2
Reviewer 2 Report
Thank you for providing your revised version.
Reviewer 3 Report
The revised version of the manuscript is satisfiying the comments/criticism. I recommend to publish the revised version of manuscript in Materials.
best wishes,